# Effect of Listening to Music on Wingate Anaerobic Test Performance. A Systematic Review and Meta-Analysis

**DOI:** 10.3390/ijerph17124564

**Published:** 2020-06-24

**Authors:** Arkaitz Castañeda-Babarro, Diego Marqués-Jiménez, Julio Calleja-González, Aitor Viribay, Patxi León-Guereño, Juan Mielgo-Ayuso

**Affiliations:** 1Health, Physical Activity and Sports Science Laboratory, Department of Physical Activity and Sports, Faculty of Psychology and Education, University of Deusto, 48007 Bilbao, Spain; arkaitz.castaneda@deusto.es (A.C.-B.); patxi.leon@deusto.es (P.L.-G.); 2Academy Department, Deportivo Alavés, 01007 Vitoria, Spain; dmarquesj@uoc.edu; 3Department of Health Sciences, Faculty of Health Sciences, Universitat Oberta de Catalunya, 08018 Barcelona, Spain; 4Laboratory of Human Performance, Faculty of Education, Sport Section, Faculty of Physical Activity and Sport, University of the Basque Country, 01007 Vitoria, Spain; julio.calleja.gonzalez@gmail.com; 5Glut4Science, Physiology, Nutrition and Sport, 01004 Vitoria-Gasteiz, Spain; aitor@glut4science.com; 6Department of Biochemistry, Molecular Biology and Physiology, Faculty of Health Sciences, University of Valladolid, 42004 Soria, Spain

**Keywords:** anaerobic, music, performance, Wingate

## Abstract

*Background*: There are many athletes who like to listen to music while making a high intensity effort. However, research into the effects of listening to music on athletic performance has provided controversial results, and it is suggested that the timing and type of music might affect the anaerobic performance response. *Purpose*: The main aim of the current systematic review and meta-analysis was to analyze the effects while listening to music tasks via the 30 s Wingate anaerobic test (WAnT) on absolute performance and relative peak power (APP and RPP), absolute and relative mean power (AMP and RMP), and fatigue index (FI). *Methods*: PRISMA guidelines were used as a basis for conducting this systematic review, with inclusion criteria being set out according to the PICOS model. Computer-based literature research was undertaken until 10 March 2020 using the following online databases: PubMed/Medline, WOS, Cochrane Library, and Scopus. The literature was reviewed with regard to the effects of listening to music on the WAnT using several music variables on: APP, RPP, AMP, RMP and FI. Hedges’ g formula was used to calculate both standard mean differences and 95% confidence intervals, in order to establish continuous outcomes. Furthermore, the I^2^ statistic evaluated systematic differences (heterogeneity) together with a random effect meta-analysis model. *Results*: This systematic review included nine articles that researched into the effects of music on WAnT performance (six studies describe improvements in APP and/or RPP, four in AMP and/or RMP and three in FI). The random effects model was used to undertake a final meta-analysis, with standardized mean differences (SMD) and magnitude of standardized mean differences (MSMD) (Hedges’ g) being pooled accordingly. The resulting meta-analysis incorporated eight studies that had been previously published, with results showing that there were no apparent beneficial effects on APP (*p* = 0.09), AMP (*p* = 0.33) and FI (*p* = 0.46) as a consequence of listening to music. However, listening to music showed beneficial effects on RPP (SMD: 0.65; 95%: CI 0.35 to 0.96; MSMD: moderate; I^2^, 0%; *p* < 0.001) and RMP (SMD: 1.03; 95%: CI, 0.63 to 1.42; MSMD: trivial; I^2^, 0%; *p* < 0.001). *Conclusion*: This systematic review and meta-analysis has shown that listening to music during the WAnT might physiologically enhance relative anaerobic exercise performance, although reasons remain speculative.

## 1. Introduction

It is extremely important for athletes to be able to accurately evaluate anaerobic power, bearing in mind that practicing a great deal of sports activities may mean that fast high-intensity exercises such as sprinting and jumping movements need to be undertaken [1]. To this end, many jumping [2], running [3] and cycling [4] tests have been proposed, and the most commonly employed protocol used to measure anaerobic components is the 30 s Wingate anaerobic test (WAnT), which is considered the anaerobic gold-standard test [5,6,7,8] in many disciplines.

Some external factors, such as music, have been related to improvements in athletic performance [9,10,11,12,13]. Generally, the benefits of music within the exercise and sports context include improved mood [14], arousal control [15], dissociation [16], and reduced perception of effort [17]. In addition, listening to music during sports activities might capture attention [18,19] and distract from fatigue and discomfort [20], given that music is able to activate the prefrontal area and minimize perceptions. Pre-task music has been successfully used as a stimulant [21], and such benefits may contribute to the ergogenic effects identified in empirical studies. These effects include improved strength [22] and improved endurance, such as in a 500 m rowing test [23] or in a 25-min submaximal cycling test [24]. However, other research found no improvement in athletes’ performance when running at 90% of maximum VO_2_ [25] or in performance in three consecutive sprints by bike [26], It has also been suggested that the timing and type of music might affect anaerobic performance response [27,28]. Specifically, a likely ergogenic effect has been evidenced by pre-task music on shorter and predominantly anaerobic tasks, including grip strength, WAnT, and short-duration sports or sports-like tasks [10,12,29].

When high levels of activation are desired, such as during high intensity exercise, the potential of musical stimulation for activation becomes very important [30]. The speed of movement and body pulses, such as heart rate and breathing rate, tend to be synchronized with the rhythm of the music. People express a preference for keeping the tempo relatively high during intense exercise [31].

Despite all the evidence to suggest that many forms of exercise and intensity have responded to music as a means for providing ergogenic assistance [26,32], there is no full consensus as to the mechanisms specifically used to enable music to enhance performance [33]. To this end, contradictory data exist as to how listening to music may affect absolute peak power (APP), relative peak power (RPP), absolute mean power (AMP), relative mean power (RMP) and fatigue index (FI) while the WAnT procedure is being performed.

Although there are few studies, as far as the author is aware, that have tended to focus on the effect of listening to music on supra-maximal anaerobic exercise [34,35], some studies have researched it during the WAnT [27,28,36,37,38,39,40,41], with contradictory results. Thus, although there are studies that did not record any significant improvement in terms of APP [28,37], the vast majority point to an improvement in PP [27,36,38,39,41,42]. On the other hand, the results obtained in the different studies give rise to more controversy regarding AMP and FI [27,28,36,37], since some studies did not record some of these data [38,41,42], while in other studies they obtained both positive and negative results with regard to the effect of music on APP, AMP and FI production during the WAnT [28,36,37,43].

Therefore, the main purpose of this systematic review and meta-analysis was to analyze the effects of listening to music on APP and AMP performance and the FI obtained using the 30 s WAnT. In addition, the data will be analyzed by taking into account the participants’ body mass (kg) in order to obtain RPP and RMP. The data were analyzed by highlighting methods and results obtained from experimental studies published in the scientific literature on the subject.

## 2. Methods

### 2.1. Literature Searching Strategies

This article is a systematic review together with a meta-analysis of existing studies, which involves researching into the effects of listening to music tasks on WAnT performance (APP, RPP, AMP, RMP and FI). Research was conducted in accordance with a preferred item subject to systematic review using meta-analysis (PRISMA) guidelines [44,45], which enabled the review’s integrity to be better enhanced, while inclusion criteria [46] were defined using the PICOS model [47] -P (Population): “adults” over 18 years old), I (Intervention): “listening to music during the WAnT”, C (Comparison): “same conditions”, O (Outcome): “APP, RPP, AMP, RMP performance and FI”, and S (Study type): “experimental studies”.

The following databases were used to carry out a structured search: PUBMED/MEDLINE, Web of Science (WOS), Cochrane library and Scopus. Proper bibliographic support was assured with these high-quality databases, and the search was completed without being confined to any specific years, with results being included up to 10th March 2020 inclusive. Search terms covered a range of Medical Subject Headings (MeSH) and free-text words for key concepts associated with both WAnT and performance, with the following unique search equation: Wingate[All Fields] AND (“music”[MeSH Terms] OR “music”[All Fields]) AND performance[All Fields]. Articles deemed to be of relevance in this sphere of activity were obtained via the snowball strategy linked to this equation. Additionally, any relevant studies were found by screening article titles and abstracts from databases and bibliographic search results, and this was then complemented by a full-text review of all articles deemed to be of potential relevance, with their adherence to inclusion criteria being subject to final analysis. Furthermore, the reference sections of all the articles found were then scrutinized, with all titles and abstracts obtained being cross-referenced in order to pinpoint any duplicates or any perceived lack of actual studies on the subject. Screening was also undertaken of titles and abstracts for full-text review at a later date. Two different authors conducted the search for previous studies separately (A.C.-B. and J.M.-A.), with any disputes being dealt with via discussion with a third-party author (D.M.-J.).

### 2.2. Inclusion and Exclusion Criteria

No filters were applied to level, gender, race or age (with participants being of legal age), so as to maximize the validity of the analysis being undertaken. Having said this, articles were ultimately selected as a result of the database search using the following inclusion criteria: (I) by establishing an experimental condition including a WAnT without music, with this then being compared to an identical experimental condition with music over the course of the test; (II) by testing the effects of listening to music on APP and/or RPP and/or AMP and/or RMP performance and/or FI obtained by the WAnT; (III) participants from the experimental group listened to any kind of music, (IV) in any language, and (V) even if no APP, RPP, AMP, RMP or FI data were provided. In addition, other inclusion criteria included studies in which both the experimental group and the control used the same resistance parameter, although it deviated from a standard of 7.5% of body mass workload. In this sense, even if the initial protocol was designed with the 7.5% of the body mass, several investigations have studied the adequacy of this measure to obtain the best performance and concluded that in non-sedentary people, the resistance provided by the bike to obtain the best performance should be greater than 7.5% [48,49,50,51]. Moreover, some studies indicated that defining resistance by the participants’ body mass is not the best parameter to apply it and it should be done taking into account the lean body mass [52].

Exclusion criteria listed as follows were also applied to the experimental protocols attached to the research: (I) studies whereby participants listened to music but were not conducted using the WAnT; (II) studies which used different outcomes other than APP, RPP, AMP, RMP and FI obtained by the WAnT; (III) the lack of a control condition; (IV) studies conducted on injured participants or on those suffering from a medical pathology; (V) studies not published in PUBMED/MEDLINE, WOS, Cochrane library or Scopus.

### 2.3. Quality Assessment

Two separate authors evaluated quality in terms of the methodology used together with any risk of bias (A.C.-B. and J.M.-A.), with any lack of consensus being subject to third-party assessment (D.M.-J.), pursuant to Cochrane Collaboration Guidelines [46]. Items on the list were broken down into seven distinct areas: random sequence generation (section bias), allocation concealment (section bias), the fact of participants and staff being blinded (performance bias), blinding of outcome assessment (detection bias), incomplete outcome data (attrition bias) and selective reporting (reporting bias), as well as other types of bias (design-specific risks of bias, baseline imbalance, blocked randomization in unblinded trials and differential diagnostic activity). In cases where criteria for low risk of bias were fulfilled, these were deemed to be “low” (plausible bias unlikely to modify results to a major extent) or “high” in cases where criteria for a high risk of bias were fulfilled (plausible bias that may reduce confidence in results to a great extent). In cases where it was unclear whether there was any risk of bias, it was deemed “unclear” accordingly (plausible bias that raises some doubts about the results).

Moreover, to determine the quality of the evidence, the authors reviewed the considered articles and provided PEDro (Physiotherapy Evidence Database) scores for each article. Only studies with PEDro scores of 4 or higher were considered for the systematic review. According to Maher et al., the PEDro scale is an 11-item scale designed for rating methodological quality of randomized control trials [53]. Each satisfied item (except for item 1) contributes one point to the total PEDro score (0–10 points) [53]. The PEDro scores were extracted from the PEDro database. If a study had not been entered into the database and scored, it was reviewed and scored by an experienced PEDro rater.

### 2.4. Outcome Variables

The literature was examined for the effects of music on WAnt performance using the following outcome variables: APP, RPP, AMP, RMP and FI.

### 2.5. Data Mining

Initially, applicable inclusion/exclusion criteria were employed for all studies, and then data pertaining to the source of study were mined separately using a spreadsheet (such data included authors and the year in which works were published, study design and type of music, sample size, bicycle resistance while the WAnT was being undertaken, participant features (gender and age), and final outcomes (APP, RPP, AMP, RMP and FI). Any disputes were then dealt with via discussion until such a time as agreement was reached.

In order to evaluate the effect of listening to music on the WAnT procedure (APP, RPP, AMP, RMP and FI), the different experiments were grouped together based on the test type selected, and eight of these studies involved gauging WAnT performance outcomes [27,28,36,37,39,41,42,43]. For meta-analysis purposes, each test or type of performance outcome was deemed to constitute a single, separate set of data and added to the relevant performance outcome. Data were mined from the tables pertaining to all the articles included so as to establish mean (M), standard deviation (SD), and sample size. Data were obtained wherever possible by contacting authors and in cases where this could not be done, the decision was made not to incorporate a particular study, bearing in mind it proved impossible to conduct the relevant statistical analysis. Any dispute was dealt with by A.C.-B. and J.M.-A. or third-party adjudication (D.M.-J.).

In the case of those studies which performed more than one consecutive WAnT, only the values of the first of them was taken into consideration, as the results of the other tests could have been influenced by fatigue. In such cases, the effect of the music during the test would not have been clear.

### 2.6. Statistical Analysis

Descriptive data of the participants’ characteristics were reported as M ± SD. Meta-analytic statistics and figures pertaining to risk of bias were compiled using Review Manager (RevMan) [Computer program]. Version 5.3. Copenhagen, Danmark: The Nordic Cochrane Centre, The Cochrane Collaboration, 2014.

The M, SD and sample size of experimental groups and their counterparts in the control group for each study were employed in order to quantify any changes in performance variables, with a view to comparing the WAnT procedure with music vs. without music. Standardized mean differences (SMD) for each study group were calculated using Hedges’ g [54], whereby mean differences were weighted by the inverse of variance in order to calculate an overall effect and its 95% confidence interval (CI). Considering that the effect of music on performance during the WAnT may vary due to other variables linked to the test or to those taking part, a decision was made to employ a random effects model in accordance with the DerSimonian and Laird method [55]. To interpret SMD magnitude (MSMD), Cohen’s criteria were used as follows: <0.2, trivial; 0.2–0.5, small; 0.5–0.8, moderate; and >0.8, large [56].

With a view to evaluating any systematic differences (heterogeneity), the I^2^ was calculated so as to prevent there being any problems arising from use of the Q statistic. The purpose of this calculation was to show the percentage of total variation noted in all studies resulting from real heterogeneity as opposed to by chance [46]. Interpreting I^2^ remains an intuitive task and ranges between 0% and 100%. A I^2^ score between 25% and 50% constitutes limited inconsistency, a I^2^ score between 50% and 75% constitutes average heterogeneity, and a I^2^ score >75% constitutes a great deal of heterogeneity [46] with a *p* < 0.05 also being applied.

## 3. Results

### 3.1. Main Search

Of the 295 articles liked to the descriptors selected that were identified in the course of the literature search, only nine fulfilled all inclusion criteria for the purpose of systematic review (Figure 1). Of these 295 articles, 28 were removed given that they were duplicates. Of the remaining 267 articles, 222 were removed after screening titles, abstracts or because the data pertaining to them were incomplete. Of the 45 full-text articles assessed for eligibility, a further 36 papers were disregarded because they were deemed to be unrelated to the effect of listening to music during the WAnT procedure (*n* = 19), or were not related to the WAnT (*n* = 17). Thus, the current systematic review included nine studies [27,28,36,37,38,39,41,42,43], while the meta-analysis included eight studies [27,28,36,37,39,41,42,43] because the Brooks and Brooks’ (2010) study failed to provide any standard deviation data [38].

### 3.2. Quality Assessment of the Experiments

The manuscripts included in this systematic review and meta-analysis showed a substantial amount of risk of bias with regard to allocation concealment and blinding of participants and personnel, as well as several studies with incomplete outcome data (Table 1 and Figure 2).

However, the nine studies obtained a high-quality methodology score (PEDro score ≥5/10), with a mean score of 5.8, according to the PEDro Scale (Table 2).

### 3.3. The Ergogenic Effect of Music on Anaerobic Performance

Participants’ characteristics, the variables measured and the protocols used in the different studies are shown in Table 3 [27,28,36,37,38,39,41,42,43]. There was a total of 267 participants (*n* = 187 men; *n* = 80 women) with a mean age of 21.9 years.

Table 4 shows the samples included in all studies involving participants in the following categories: students (*n* = 6) and physically active (*n* = 2). Bicycle resistance during the test did not account for 7.5% of body mass in two out of eight studies. In one of them, it was 0.090 kp/kg of body mass, and in the other, the resistance used was not specified. On the other hand, the type of music used in the studies was selected by the participants freely or by having to fulfil some requirement (*n* = 4), or music was selected by the researchers owing to its rhythm or characteristics (*n* = 3). No inclusion criteria were applied to the type of music, and so both low and high tempo music were accepted.

Moreover, the anaerobic capacity parameters researched can be expressed both in absolute (W) and in relative values (W/kg) [57,58]. Some of the studies included in this systematic review and meta-analysis show results in absolute values, whereas others report data in relative values. Consequently, the decision was made to perform five meta-analyses, based on APP, RPP, AMP, RMP and FI. It was thought this could help to understand the effect of music on anaerobic capacity.

### 3.4. Effect on Peak Power Meta-Analysis

The effect of music on APP performance during the WAnT procedure is shown in Figure 3. Although all the articles included in this section evidenced the fact that music could lead to an improvement in the experimental group versus the control group, the results obtained from the meta-analysis indicated that listening to music did not statistically improve absolute APP (SMD:, 0.20; 95% CI: −0.03 to 0.43; MSMD, small; I^2^, 0%; *p* = 0.09).

Figure 4 shows the overall effect of music on RPP performance during the WAnT. In this case, the results indicate that listening to music leads to improvements in RPP (SMD: 0.65; 95%: CI 0.35 to 0.96; MSMD: moderate; I^2^, 0%; *p* = 0.0001).

### 3.5. Effect on Mean Power Meta-Analysis

The effect of music on AMP performance during the WAnT procedure is shown in Figure 5. Albeit with the exception of the article by Pujol et al., which showed that listening to music during the WAnT does not favor AMP, the rest of the articles included in this section tended to show improvements in the experimental group. The results of the meta-analysis indicate that listening to music did not statistically improve absolute AMP during the WAnT (SMD: 0.11; 95%: CI, −0.11 to 0.34; MSMD: trivial; I^2^, 0%; *p* = 0.33).

Figure 6 shows that listening to music significantly improves RMP performance during the WAnT (SMD: 1.03; 95%: CI, 0.63 to 1.42; MSMD: trivial; I2, 0%; *p* = 0.00001).

### 3.6. Effect on Fatigue Index Meta-Analysis

Figure 7 shows that no significant improvements were found in the FI of the WAnT with and without music (SMD, −0.09; 95% CI, −0.34 to 0.15; MSMD, trivial; I^2^, 28%; *p* = 0.46).

## 4. Discussion

A summary of the effects of listening to music on WAnT performance tests linked to APP, RPP, AMP, RMP and FI constituted the chief aim of this systematic review and meta-analysis. The main results indicate that listening to music did not evidence any statistical improvements in FI. Moreover, when the athletes listened to music during the WAnT, the results obtained from the meta-analysis show no statistical difference when interpreting whether listening to music improved APP and AMP. However, the results show that listening to music during the WAnT statistically improved RPP and RMP.

### 4.1. Effect of Music on Peak Power and Mean Power Performance

The results obtained from this systematic review and meta-analysis (only nine articles according to our updated knowledge) suggest that listening to music may have a small effect on APP improvements and a trivial effect on increasing AMP. However, when the parameters were expressed in relation to body mass (RPP and RMP), the effects were moderate. Although some psycho-physiological factors might explain these effects (listening to different genres of music induces different psycho-physiological responses) [59], one explanatory variable of the variance in APP-RPP and MP-AMP performance might be the difference in studies included in each analysis, especially regarding the type of music. Thus, recent studies have concluded that listening to music would enhance the speed of performance (related to anaerobic performance). In addition, music with a higher tempo (example of Brohmer et al. 2006, with AC/DC’s music) increased the speed more [60], and in this sense, only Isik et al. 2015 provided data regarding all power output outcomes (APP, RPP, AMP and RMP) [27]. These results are consistent with other studies that have measured PP using other types of test such as vertical jump, in which music improves some of the parameters related to force in the vertical test [61].

The meta-analysis performed by subgroups with APP, MPA, RPP and RMP should help to demonstrate that there is consistency among the results obtained from the subgroups (APP and RPP as well as MPA and RMP). However, and given that body mass is inherent in the body itself, the results obtained from the APP and MPA meta-analyses should be consistent with the results of RPP and RMP. Given these significant findings from this subset of studies, and the fact that the effect becomes non-significant (albeit almost significant in the PPP analysis) as more studies become included, this suggests that perhaps either the variability added by the inclusion of more studies reduces any potential “real” effect of the music intervention, or that the musical intervention does not in fact exert a true effect. However, it is important to take into account the low heterogeneity (0%) of the sample, and so the test is considered to have been conducted under similar conditions with similar subjects—in other words, the only difference between studies is their power to detect the outcome of interest [62].

The way in which music has an influence on mood, emotion, feeling (i.e., feelings of pleasure or displeasure), cognition (thought processes) and behavior forms part of its psychological effects on exercise. Consequently, and throughout the three routes for emotion induction—memory, empathy and appraisal [63]—emotional response to music may also positively affect WAnT performance. It has been previously pointed out that, in the presence of music, athletes reported increased task motivation and more positive feeling during the WAnT [36]. Indeed, motivational music contains performance components that may enhance certain physiological and psychological factors, as well as possibly also physical performance [43], while certain psycho-physiological factors such as will power, psychological balance and the autonomous nervous system may be the root cause of such an effect, [27] although they were not evaluated by researchers. It could therefore be assumed that the emotional response to music may be responsible for the trivial and small improvements in APP and AMP, respectively, although this cannot be the explanatory variable for the variance in absolute and relative values of PP and MP.

The purpose of selecting music is, generally speaking, to ensure the target set out by the individual is optimized [64], hence a major role can also be played in establishing the psycho-physiological effect of music on physical performance capacity in terms of the kind of music selected by that individual [27]. Indeed, participants may sense “segmentation,” in the moments prior to doing exercise, whereby music segments which are to be played are anticipated by the subject, resulting in an enhanced state of arousal and a conscious trigger for increased work output [10,65,66]. Consequently, music selection and a plausible “segmentation” could be important factors in explaining the anaerobic performance improvements during the WAnT. However, both factors cannot be responsible for differences observed between absolute and relative values, both in PP and MP, due to the fact that a similar number of studies included in this meta-analysis made it easy for participants to select the type of music, whereas in others, types of music were imposed by researchers.

Another factor related to the psycho-physiological effects of music, which may have influenced these results, is music tempo. In a recent review, up-tempo music was thought to enhance performance, whereas it was considered that slow-tempo music could lead to either calming or negative reactions [67]. However, our meta-analytical study included studies of music in both tempos: some with music in a tempo that was very fast and upbeat, and some with soft, slow, relaxing music.

In keeping with the aforementioned data involving PP and MP, it is suggested that music might have an ergogenic effect on anaerobic performance during the WAnT due to different mechanisms, which must be confirmed in the future. Considering SMD values and statistical significance, these effects are greater in PP (Figure 4 and Figure 5) than in MP (Figure 6 and Figure 7).

### 4.2. Effect on Fatigue Index Performance

The FI refers to the reduction in power over the course of the test, and may be expressed as the difference between the peak and minimal power divided by PP. Consequently, it can be viewed as a percentage of PP [57,68]. Although it is directly related to the above-mentioned performance rates obtained via the WAnT, the FI was considered to be less reliable than the other two WAnt rates and, bearing in mind it mainly depends on aerobic performance, the extent to which it was deemed valid was open to debate [8].

It is important to take into account that the FI is very sensitive to training effects and to the specialist sport practiced by the subject [69], along with depending on the type of participant muscle fibers, since a higher percentage of muscle fibers facilitates a higher PP but with a higher FI. This is due to the difficulty in maintaining those high power values with a lower aerobic capacity of the fast fibers [70].

In spite of the fact that the meta-analysis of FI was carried out on most of the studies included in the systematic review, the results show that music does not have any ergogenic effect on FI. It is difficult to understand why this variable does not benefit from it, since the exact mechanism by which music may improve PP and MP is still unknown [10]. The main explanatory hypothesis that has been proposed is the following: although music could have a great effect on the initial burst of power (effect on PP, and overall MP), and may provide an extra impetus to participants to be motivated or energized to make a strong start with the WAnT, this effect could diminish as the test progresses [36].

### 4.3. Strengths, Limitations and Future Lines of Research

Individual trials in this systematic review may seem to measure the same outcome but may also produce results that are not consistent with each other [71], insofar as differences may be noted among studies in assessing the extent to which they are treated or exposed. Random sampling error may be responsible for some variance among studies, although their heterogeneous nature may be the cause in other cases. Heterogeneity may derive from quite a few sources, such as differences in treatment and the actual population subject to treatment, as well as study design and the method used for data analysis [72]. In this case, sources of heterogeneity can be related to the number of participants of one gender or another, implementation of the protocol, or other variables such as warm-up, time of day when the tests were carried out, etc. [73,74]. Making any specific assertions about heterogeneity also tends not to be straightforward, given that meta-analyses—including the one carried out here—mostly tend to be limited [72].

Taking this information into consideration, several factors may have contributed to the limited amount of inconsistency obtained in APP (0%), RPP (0%), AMP (0%), RMP (0%) and FI (28%): all subjects were of a similar age range, between 18 and 38 years (the vast majority of them being close to 20 years), the sample comprised healthy subjects or active people, gender differences in the sample were not great enough to give rise to detectable changes, and also in the vast majority of studies (six out of eigh), the WAnT was performed according to the standardized protocol.

One of the most important limitations of this meta-analysis (nine articles), albeit bearing in mind the updated state of the art, is the fact that the data were found in different ways, with some of the records provided being relative, absolute or both, which makes it difficult to draw conclusions. Although the calculations of the meta-analysis were made according to sub-groups (main limitation) with the aim of clarifying the different tendencies, the way in which the data were found in each article determined how that data were grouped, and therefore it has not been possible to obtain unique and definitive meta-analytical calculations for each variable measured.

The fact that few studies were found that focus on how music is applied while the WAnT procedure is performed limited the scope of this systematic review and meta-analysis (nine studies in the case of the systematic review and eight in the case of the meta-analysis). As a result, data obtained from both sexes were analyzed accordingly, with different competitive levels and research protocols, and studies were put together that applied different types of music. Therefore, it should be noted that neither gender nor the type of music applied was taken into account. In fact, there are mixed studies [37,43], in which fast rhythm and slow rhythm music are applied, which may have influenced the results [37,43]. Comparing the results obtained among subgroups (level of participants, type of music…) would have enriched the conclusions drawn from the review, but due to the small number of articles found, it is impossible to make a correct comparison. In addition, it is important to mention the small sample size of the studies selected, because none of the studies in the meta-analysis exceeded 30 participants, thus reducing the potential of the meta-analysis.

In studies in which more than one test was performed on the same day [28,42], only the values of the first test were taken into account, so that fatigue would not influence the results. Anyway, other variables, including some not specified by the researchers, may influence the results of the research. Those related to WAnT include: the type of warm-up before the test [75] as a warm-up with changes in pace that may help to improve performance later on in the test; resistance provided by the cicloergometer [76] as a type of resistance that is too low, such as the recommended 7.5% of body mass, may not help to achieve maximum performance; the time of day when the test is done [74], i.e., in the evening, may increase test performance; the change in position adopted by the subject in the cicloergometer during the test [77] may help to improve performance; the geometry of the bicycle [78] which, although it would seem to have no influence, has not as yet been subject to much study. As for the music used: the volume at which the music is played can influence performance [26], and a higher volume can be more motivating; the participants’ previous experiences with music [10], because a song can recall moments and emotions for one subject but not for another; whether the music is their favorite or not [26,32], which can provide motivation; and generally speaking, their level of education (in six of the eight research cases, they were students, but level of education was not provided) [10]. Moreover, although all the researchers played the music during the test [36], some of them also played it 20″ (even earlier) before the test [41], while others did so at the start of the test [28]. Given that it is a 30 s test, the first part of the song (the chorus, for example) that is heard during the test can also have an influence [10].

Future research projects should confirm the results obtained during the course of this review and meta-analysis, and include other types of population, different types of music and volume of the music, as well as help clarify the unclear reasons why music improves anaerobic performance.

### 4.4. Practical Applications

Anaerobic power is of great importance in many types of sport [79], so any help in improving performance in this regard should be considered through the development of a training program. Until now, music has been used as a motivational or relaxing element, albeit probably without giving the importance it deserves to such a simple variable that may have an ergogenic effect. It is important to take into account that this possible effect may not always interest sportsmen, since if what is being sought is to maintain great power that is as constant as possible (a low FI), the music may not interest us, whereas if the objective is a high PP value, then it seems that its possible effect might prove interesting [12]. However, while music may be of interest for increasing PP, based on current meta-analysis, there is insufficient evidence to provide a recommendation.

There are some variables that can influence the response of the subject in order for them to improve their performance of the test with music: sex, age, personality type, frequency of exercise, level of physical fitness, type of attention, exercise environment, etc. That is why we think it would be appropriate to continue researching into the ergogenic effect of music on the WAnT, albeit taking into account these variables.

## 5. Conclusions

Based on the results obtained from this systematic review and meta-analysis, it is clear that listening to music during the WAnT does not improve anerobic performance in adults over 18 years old. Although there is a tendency to improve on results obtained in most studies, these differences are not statistically significant. To the best of our knowledge, there have been scant studies published on this topic and future research might confirm these findings. Although the results reported in this systematic review and meta-analysis show a trend towards improvement, we cannot confirm this yet, and although this is still subject to speculation, it is believed that the effects of music might derive from other unknown psychophysiological factors.

## Figures and Tables

**Figure 1 ijerph-17-04564-f001:**
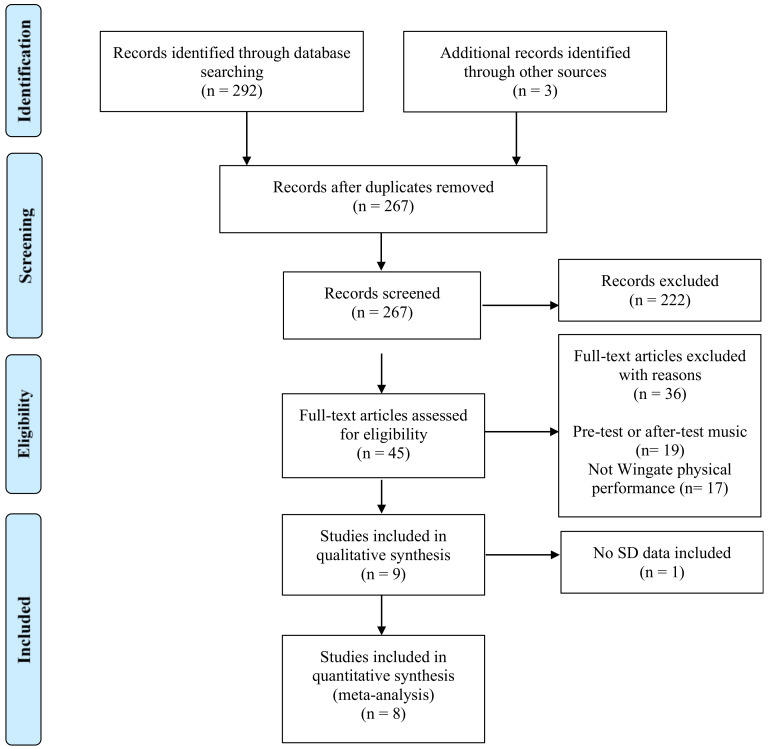
Flow chart of study selection. SD: Standard deviation.

**Figure 2 ijerph-17-04564-f002:**
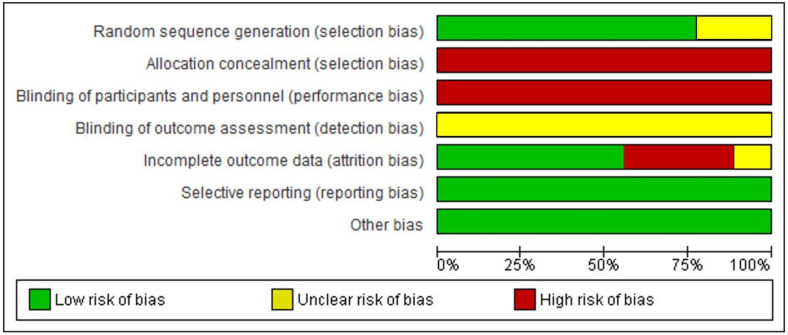
Graph showing risk of bias: percentages are provided for all studies included, based on a review of authors’ opinions about each risk of bias item.

**Figure 3 ijerph-17-04564-f003:**
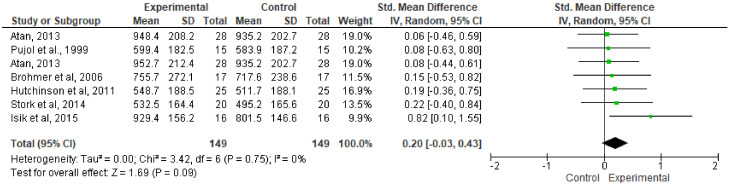
Forest plot comparing the effects of music on absolute peak power performance.

**Figure 4 ijerph-17-04564-f004:**
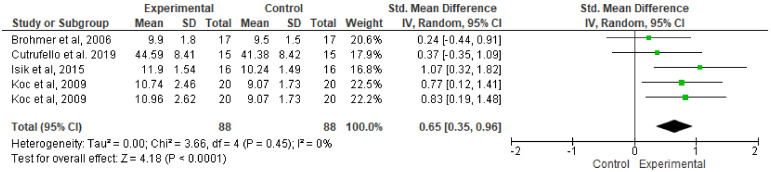
Forest plot comparing the effects of music on relative peak power performance.

**Figure 5 ijerph-17-04564-f005:**
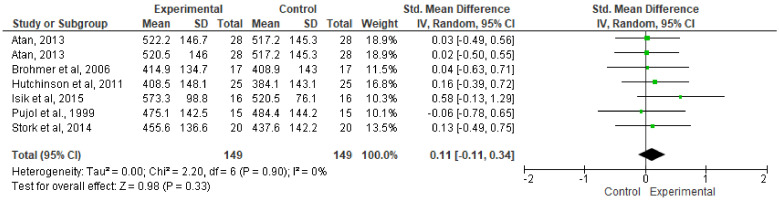
Forest plot comparing the effects of music on absolute mean power performance.

**Figure 6 ijerph-17-04564-f006:**
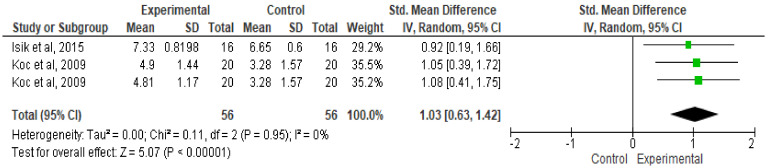
Forest plot comparing the effects of music on relative mean power performance.

**Figure 7 ijerph-17-04564-f007:**
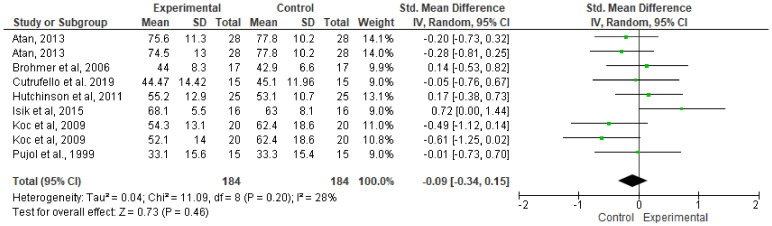
Effects of music on FI performance contrasted using forest plot.

**Table 1 ijerph-17-04564-t001:** Summary showing risk of bias: all studies included are subject to review of authors’ opinions about each risk of bias item. 
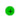
 shows low risk of bias; 
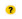
 shows unknown risk of bias; 
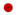
 shows high risk of bias.

Authors	Random Sequence Generation (Selection Bias)	Allocation Concealment (Selection Bias)	Blinding of Participants and Personnel (Performance Bias)	Blinding of Outcome Assessment (Detection Bias)	Incomplete Outcome Data (Attrition Bias)	Selective Reporting (Reporting Bias)	Other Bias
Atan., 2013 [37]	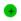	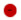	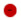	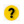	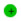	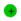	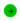
Brohmer et al., 2006 [39]	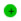	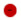	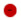	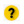	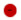	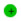	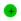
Cutrufello et al., 2019 [41]	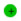	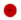	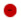	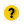	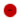	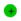	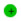
Hutchinson et al., 2011 [36]	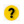	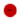	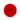	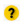	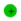	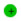	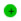
Isik et al., 2019 [27]	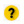	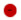	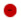	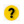	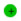	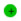	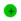
Koc et al., 2010 [43]	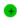	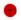	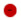	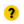	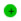	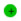	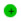
Pujol et al., 2006 [28]	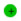	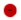	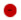	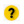	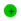	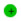	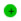
Stork et al., 2014 [42]	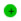	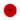	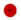	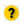	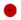	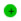	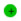
Brooks et al., 2010 [38]	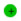	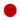	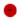	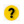	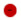	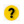	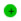

**Table 2 ijerph-17-04564-t002:** Quality assessment with the PEDro scale.

Article	Items by Number on the PEDro Scale											Total Score
	1	2	3	4	5	6	7	8	9	10	11	
Atan., 2013 [37]	Y	Y	N	Y	N	N	N	Y	Y	Y	Y	6
Brohmer et al., 2006 [39]	Y	Y	N	Y	N	N	N	Y	Y	Y	Y	6
Cutrufello et al., 2019 [41]	Y	Y	N	Y	N	N	N	Y	Y	Y	Y	6
Hutchinson et al., 2011 [36]	Y	N	N	Y	N	N	N	Y	Y	Y	Y	6
Isik et al., 2019 [27]	Y	N	N	Y	N	N	N	Y	Y	Y	Y	5
Koc et al., 2010 [43]	Y	Y	N	Y	N	N	N	Y	Y	Y	Y	6
Pujol et al., 2006 [28]	Y	Y	N	Y	N	N	N	Y	Y	Y	Y	6
Stork et al., 2014 [42]	Y	Y	N	Y	N	N	N	Y	Y	Y	Y	6
Brooks et al., 2010 [38]	Y	Y	N	Y	N	N	N	Y	Y	Y	Y	6

N, criterion not fulfilled; Y, criterion fulfilled; 1, eligibility criteria were specified; 2, subjects were randomly allocated to groups or to a treatment order; 3, allocation was concealed; 4, the groups were similar at baseline; 5, all subjects were blinded; 6, all therapists were blinded; 7, all assessors were blinded; 8, measures of at least one key outcome were obtained from over 85% of the subjects who were initially allocated to groups; 9, intention-to-treat analysis was performed on all subjects who received the treatment or control condition as allocated; 10, the results of between-group statistical comparisons are reported for at least one key outcome; 11, the study provides both point measures and measures of variability for at least one key outcome; total score, each satisfied item (except the first) contributes 1 point to the total score, yielding a PEDro scale score that can range from 0 to 10.

**Table 3 ijerph-17-04564-t003:** Studies included in the systematic review: features of those taking part and of the relevant interventions.

Level of participants	Physical education students	3 studies [27,37,41]
	Physically active	2 studies [36,42]
	Moderate to high fitness	3 studies [28,39,43]
	Low risk volunteers	1 study [38]
Kind of music	120 beats/min o more (fast rhythm)	3 studies [30,31,37]
	80 beats/min approx. (slow music)	1 study [39]
	Motivational music	3 studies [36,38,41]
	2 types of music on two different days (1 day 120 beats/min or more and another day 80 beats/min approx.	2 studies [35,40]
Resistance applied during the WAnT	7.5% of body mass in kg	6 studies [27,28,36,37,42,43]
	Undisclosed	2 studies [38,39]
	0.090kp/ kg of body mass	1 study [41]
Use of cleats	Undisclosed	8 studies [27,36,38,39,41,42,43]
	Using cleats	1 study [37]
Experience or previous practice during the test	Undisclosed	5 studies [28,36,38,39,43]
	They tried the test days before	4 studies [27,37,41,42]
Biomechanical aspects during the test	Undisclosed	8 studies [27,28,36,38,39,41,42,43]
	Bicycle dimensions adjusted to the participants and the whole test sitting on the bike	1 study [37]
Warm Up	Undisclosed	3 studies [28,38,42]
	5–10 min without sprints	2 studies [39,43]
	5–10 min with sprints	3 studies [27,36,37]
	3 min warm up	1 study [41]

**Table 4 ijerph-17-04564-t004:** Breakdown of studies selected for systematic review.

Author/s-Year	Population	Intervention	Outcomes Analyzed	Main Conclusions
Atan, 2013 [37]	28 males 21.26 ± 1.86 years Physical Education students	Fast rhythm music (‘Viva La Van’ at 200 beats/min) Slow rhythm music (‘First Born’s Lullaby’ at 70 beat/min) Resistance setting: 7.5% body mass	APP AMP FI	↔ ↔ ↔
Brohmer et al., 2006 [39]	17 (8 males, 9 females) 21.2 ± 0.7 years College students—physically fit	Thunderstruck by AC/DC (85–90 beats/min) Resistance setting: non-defined	APP RPP FI	↑ ↑ ↔
Brooks et al., 2010 [38]	63 (24 males, 39 females) 23.5 years males, 21.5 years females Low-risk volunteers	Self-selected motivational music Resistance setting: non-defined	APP RPP AMP RMP FI	↑ ↑ ↑ ↑ ↑
Cutrufello et al., 2019 [41]	15 (8 males, 7 females) (20.1 ± 1.79 years) Healthy, college-aged students	Their selected songs Resistance setting: 0.090 kp/ kg of body mass	RPP FI	↑ ↔
Hutchinson et al., 2011 [36]	25 (13 males and 12 females) 20.8 ± 5.4 years Physically active	Sandstorm by Darude Resistance setting: 7.5% body mass	APP AMP FI	↑ ↑ ↔
Isik et al., 2015 [27]	16 males 23.19 ± 3.02 years Physical Education students	Self-selected music from 50 popular songs at 120–130 beats/min Resistance setting: 7.5% body mass	APP RPP AMP RMP FI	↑ ↑ ↑ ↑ ↑
Koc et al., 2009 [43]	20 (14 males, 6 females) 19.97 ± 11.34 years College students—physically fit	Slow music and fast music (beats/min non-defined) Resistance setting: 7.5% body mass	RPP RMP FI	↑ ↑ ↑
Pujol et al., 1999 [28]	15 (12 males, 3 females) 24.0 ± 3.4 years College students—moderate to high fitness	Their favorite type of music at 120 beats/min Resistance setting: 7.5% body mass	APP AMP FI	↔ ↔ ↔
Stork et al., 2014 [42]	20 healthy and moderately active, 10 males and 10 females (22.5 ± 4.3 years)	Self-selected music (80dB) Resistance setting: 7.5% body mass	APP AMP FI	↑ ↑ -

APP: absolute peak power; RPP: relative peak power; AMP: absolute mean power; RMP: relative mean power; FI: fatigue index. ↑ Statistically significant increase (*p* < 0.05); ↔ no statistical significance changes (*p* > 0.05); ↓: statistically significant decrease (*p* < 0.05).

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
