# Peer review of "Effect of Listening to Music on Wingate Anaerobic Test Performance. A Systematic Review and Meta-Analysis"

_ijerph, 2020, doi:10.3390/ijerph17124564_

Round 1

Reviewer 1 Report

Thanks for your contribution.

After reading and reviewing the manuscript, I add my contributions to improve this article:

Have you registered the review protocol and obtained a code from the International Prospective Register of Systematic Reviews (PROSPERO)?

Has a search been performed including the PROSPERO database?

In the Inclusion and Exclusion criteria section, on line 137, what do you mean by "... with a medical condition, illness or injury." What kind of diagnoses would they be? Or is it only for healthy patients?

On line 172, you explain that you have included other types of bias, what are those different types?

In this same section, Table 1 and Figure 1 have been included, these two tables explain the degrees of bias prior to performing the meta-analysis, so they are results and should be in the Results section.

Check the readability and consistency of the wording between lines 204-206.

Results:

In "Figure 2. Flow chart of study selection" the studies that met the inclusion and exclusion criteria for their qualitative analysis were 9, and in results you describe that the articles that met the inclusion criteria (line 202) were 8.

Review Your manuscript, so that there is coherence between the results obtained by the systematic review and the studies used for its quantitative analysis, are related procedures, although their performance and results must be well explained.

In Table 2, there are data that do not coincide or are not clear. In "Use of cleats", the study by Pujol TJ et al. it is not explained whether he uses cleats, does not use cleats, or does not specify his use. In "Kind of music" there are included studies (Atan T .; Koc et al.) In two sections, a fourth section should be added specifying it.

In Table 3, 8 studies are included. This table corresponds to the extraction of data from the systematic review (qualitative synthesis), so it must have 9 studies or the data presented in the text are incorrect.

Line 302, check the errors in the text.

Discussion:

Line 334. The hypothesis described is yours or from the study carried out by Hutchinson J.C. et al.? It is not clear.

There are flaws in the coherence of the given data.

There is also a structure failure, between the Methods and the Results. There is a Methodology failure if there is no record of the systematic review in PROPERO.

Thanks for your contribution, it is an interesting study although it needs to be improved for publication.

Author Response

Point-by-Point Response to Reviewer’s Comments

We would like to sincerely thank the reviewers for their helpful recommendations. We have seriously considered all the comments and carefully revised the manuscript accordingly. Revisions are highlighted in yellow through the manuscript to indicate where changes have taken place. We feel that the quality of the manuscript has been significantly improved with these modifications and improvements based on the reviewers’ suggestions and comments. We hope our revision will lead to an acceptance of our manuscript for publication in International Journal of Environmental Research and Public Health.

In advance,

Kind regards

REVIEWER 1

REVIEWER: Have you registered the review protocol and obtained a code from the International Prospective Register of Systematic Reviews (PROSPERO)?

AUTHORS: Thank you for your interest. We had not made yet the record of the review in PROSPERO because they recommend in the second step uploading the latest version, once reviewed, in order to provide the most recent version:

“Step 2: Ensure that your review protocol is in its (near) final form and that no major changes are anticipated at this stage - e.g. if your protocol will be peer reviewed it will usually be sensible to wait until this is complete before registering.”

On the other hand, the PROSPERO database at this time is unable to accept our application given that the outcomes that we analysed were on athletic performance related and they are not on human health related:  

“Does this review have at least one outcome directly related to human health or is it a methodology review that has a clear link to human health?

Sorry - your protocol is not eligible for inclusion in PROSPERO. We hope that this screening process has saved you the time that might be spent preparing a submission that would be rejected and that you will register future systematic reviews in PROSPERO. Further information about PROSPERO scope and eligibility can be found in our guidance document.”

REVIEWER: Has a search been performed including the PROSPERO database?

AUTHORS: Thank you for your interest. We carried out a search using the key words used in our review and have not found any articles related to our topic.

REVIEWER: In the Inclusion and Exclusion criteria section, on line 137, what do you mean by "... with a medical condition, illness or injury." What kind of diagnoses would they be? Or is it only for healthy patients?

AUTHORS: Thank you for your interest. This sentence refers to any research conducted on the effect of music on Wingate test performance in injured or pathological populations. In order to clarify this issue, the authors have included in the exclusion criteria section next sentence: and (IV) studies conducted with injured or with medical pathology participants.

REVIEWER: On line 172, you explain that you have included other types of bias, what are those different types?

AUTHORS: Thank you for your observation. Following Cochrane handbook other types of bias are as well as other types of bias (design-specific risks of bias, baseline imbalance, blocked randomization in unblinded trials and differential diagnostic activity). In this sense, the authors have included this sentence in Quality assessment of the experiments section.

REVIEWER: In this same section, Table 1 and Figure 1 have been included, these two tables explain the degrees of bias prior to performing the meta-analysis, so they are results and should be in the Results section.

AUTHORS: Thank you for your recommendation. We have made the change and have placed both figure 1 and table 1 in the results section. Figure 1 is now called figure 2.

REVIEWER: Check the readability and consistency of the wording between lines 204-206.

AUTHORS: Thank you for your recommendation. We have made the correction in the text, and with de changes made the line has become 181: “Of the 45 full-text articles assessed for eligibility, a further 36 papers were disregarded because they were deemed to be unrelated to the effect of listening to music during the WAnT procedure (n=19), or were not related to the WAnT (n=17).”

Results:

REVIEWER: In "Figure 2. Flow chart of study selection" the studies that met the inclusion and exclusion criteria for their qualitative analysis were 9, and in results you describe that the articles that met the inclusion criteria (line 202) were 8.

AUTHORS: Thank you for your observation. We have made corrections to the articles included and with de changes made the line has become 179: Of the 295 articles liked to the descriptors selected that were identified in the course of the literature search, only 9 fulfilled all inclusion criteria.

REVIEWER: Review Your manuscript, so that there is coherence between the results obtained by the systematic review and the studies used for its quantitative analysis, are related procedures, although their performance and results must be well explained.

AUTHORS: Thank you for your observation. Figure 1 show that 1 of the studies included in systematic review did not present standard deviation data necessary to perform meta-analysis statistics. For clarify this issue in the text we have added next sentence in line 186: “review included 9 studies [30,31,34–40], while the meta-analysis included 8 studies [30,31,34,35,37–40] because Brooks and Brooks (2010) study did no present standard deviation data [36].”

REVIEWER: In Table 2, there are data that do not coincide or are not clear. In "Use of cleats", the study by Pujol TJ et al. it is not explained whether he uses cleats, does not use cleats, or does not specify his use. In "Kind of music" there are included studies (Atan T .; Koc et al.) In two sections, a fourth section should be added specifying it.

AUTHORS: Thank you for your observation. We have corrected this mistake:

  • Undisclosed 8 studies [30,34,36–40] and
  • Using cleats 1 study [35]

Regarding “Kind of music””, we have added another row indicating 2 types of music in two different days (1 day 120 beats/min o more and other day 80 beats/min approx.) for Atan T.; and Koc et al. studies.

REVIEWER: In Table 3, 8 studies are included. This table corresponds to the extraction of data from the systematic review (qualitative synthesis), so it must have 9 studies or the data presented in the text are incorrect.

AUTHORS: Thank you for your observation. We have added Brooks et al. 2010 study in table 3.

REVIEWER: Line 302, check the errors in the text.

AUTHORS: Thank you for your recommendation. We Solved them (now in line 291)-

Discussion:

REVIEWER: Line 334. The hypothesis described is yours or from the study carried out by Hutchinson J.C. et al.? It is not clear.

AUTHORS: Thank you for your interest. The given hypothesis is a Hutchinson J.C. et al. hypothesis

REVIEWER: There are flaws in the coherence of the given data.

AUTHORS: Thank you for your opinion. The authors have tried to correct flaws in the coherence of the given data.

REVIEWER: There is also a structure failure, between the Methods and the Results. There is a Methodology failure if there is no record of the systematic review in PROPERO.

AUTHORS: Thank you for your indication. As the authors have explained in previous points and following PROSPERO database indications, we can´t submit this review in PROSPERO. However, although this systematic review and meta-analysis are not registered in PROSPERO, we have followed the PRISMA protocol and the same structure that previous systematic review and meta-analysis published recently by or research group in other Q1 journal without PROSPERO registration.

Reviewer 2 Report

The authors carried out a meta-analysis to verify whether listening to music during the exercise improved sports performance in the 30s Wingate test, showing no strong correlation.

As suggested in the text, the physical response to music depends on an athlete's interindividual factors such as the music-specific interest, the type of music the athlete prefers, a piece of memory music can arouse and the motivational boost it can give.

At the same time, even a friend who encourages the athlete during the test can have similar motivational effects as happens with personal trainers who encourage performance.

Given the great variability of influencing factors and the subjectivity of the context, I do not think this type of analysis is scientifically interesting also because the works involved in the meta-analysis are different and with many biases without a robust correlation.

Furthermore, the text is full of grammatical errors and stylistic errors and must be completely revised by an English native speaker or through grammar correction software. 

In order to reconsider the paper suitable for publication it must be completely revised grammatically and by adding more evidence on the effect of music in sports performances although studies to date are scarce and highly controversial.

Author Response

Point-by-Point Response to Reviewer’s Comments

We would like to sincerely thank the reviewers for their helpful recommendations. We have seriously considered all the comments and carefully revised the manuscript accordingly. Revisions are highlighted in yellow through the manuscript to indicate where changes have taken place. We feel that the quality of the manuscript has been significantly improved with these modifications and improvements based on the reviewers’ suggestions and comments. We hope our revision will lead to an acceptance of our manuscript for publication in International Journal of Environmental Research and Public Health.

In advance,

Kind regards

REVIEWER 2

REVIEWER: As suggested in the text, the physical response to music depends on an athlete's interindividual factors such as the music-specific interest, the type of music the athlete prefers, a piece of memory music can arouse and the motivational boost it can give.

At the same time, even a friend who encourages the athlete during the test can have similar motivational effects as happens with personal trainers who encourage performance.

Given the great variability of influencing factors and the subjectivity of the context, I do not think this type of analysis is scientifically interesting also because the works involved in the meta-analysis are different and with many biases without a robust correlation.

Furthermore, the text is full of grammatical errors and stylistic errors and must be completely revised by an English native speaker or through grammar correction software. 

In order to reconsider the paper suitable for publication it must be completely revised grammatically and by adding more evidence on the effect of music in sports performances although studies to date are scarce and highly controversial.

AUTHORS: Thank you for your recommendation. As we have commented in previous point, the manuscript was reviewed grammatically by an English native.

When carrying out a meta-analysis, a number of articles with common characteristics are grouped together to try to give a conclusion about all of them, assuming that they have not carried out exactly the same methodology when conducting the research. These methodological differences are what may produce different results. As you said, music can produce different effects in each individual, but as the selected researches do not value the effect of music in each individual but in a group. In this meta-analysis we only try to draw a conclusion about which effect music can have (without specifying which type), in the performance of a Wingate test (without taking into account many test variables that are applied differently in each research) in a generic way, and without ignoring that these effects may not occur in the same way in all subjects, as it happens with most of the conclusions obtained in the great majority of researches. 

This is not the first time that a review has been published on the effect of music on sports performance, (Smirmaul BP. Effect of pre-task music on sports or exercise performance. J Sports Med Phys Fitness. 2017;57(7-8):976‐984. doi:10.23736/S0022-4707.16.06411-2 ) and we believe that since music is such a common variable and is used both in physical activity in general and in research, a review on this subject could be interesting.

We are aware, and we have reflected this in the article, that it would have been much more interesting to be able to make comparisons of subgroups (by gender, type of music, resistance applied to the bike...) but due to the small number of articles found we have not been able to do so. Even so, we understand that 8-9 articles are a sufficient number to be able to carry out a review with meta-analysis, since there are meta-analyses published with the same number even with less:

Serraglio CR, Zanella L, Dalla-Vecchia KB, Rodrigues-Junior SA. Efficacy and safety of over-the-counter whitening strips as compared to home-whitening with 10 % carbamide peroxide gel--systematic review of RCTs and metanalysis. Clin Oral Investig. 2016;20(1):1‐14. doi:10.1007/s00784-015-1547-8

Micali LR, Matteucci F, Parise O, et al. Clinical outcomes of automated anastomotic devices: A metanalysis. J Card Surg. 2019;34(11):1297‐1304. doi:10.1111/jocs.14186

De Rango P, Cao P, Parlani G, Verzini F, Brambilla D. Outcome after endografting in small and large abdominal aortic aneurysms: a metanalysis. Eur J Vasc Endovasc Surg. 2008;35(2):162‐172. doi:10.1016/j.ejvs.2007.10.015

Although we have effectively selected few studies, and there are differences in the protocols used between the studies (use of coves, type of music...) all of them measure the possible improvement of performance in the same variables (PP, MP and FI) by modifying a single variable, listening to music or not listening to music. Although the methodologies are not identical between studies, they are intra-study, which we believe makes possible to do a review and metanalysis of this subject.

Reviewer 3 Report

In this manuscript, the authors conducted a systematic review and meta-analysis of the effect of listening to music during the Wingate anaerobic test on the test performance. They found that listening to music enhanced relative peak power and relative mean power. The main analysis and results are concise and informative. It provides important meta-analytic evidence on the benefits of music in the exercise and sports literature.

One of my major concerns is the typos and double spaces throughout the manuscript (e.g., the first RMP in the abstract should actually be RPP), some major statistic results seem also wrongly presented (e.g., p=0.033 vs 0.33). All these reduce the credibility of the paper and compromise fluent reading.

My second major concern is that the authors did not perform any subgroup analyses. This was perhaps because the heterogeneity analyses were not significant. However, even in the absence of heterogeneity within trials, subgroup analysis may provide important insights (see https://www.ncbi.nlm.nih.gov/pmc/articles/PMC2900281/; https://cegh.net/article/S2213-3984(18)30099-X/pdf). The authors may reconsider the differences between studies shown in, for instance, Figure 5 & 8, and check whether those differences could actually be attributed to subgroup differences. Relevant to this point, the authors mentioned some additional possible subgroup differences in lines 363-371, so the information of these features could not be included in Tables 2 & 3? Is there any possibility that some of these may cause any not-yet-discovered subgroup differences?

Another concern is that there seem to be studies that investigated the effect of listening to music using paradigms other than WAnT, since the authors mentioned that they excluded these studies (no problem at all); I think they may discuss the current findings of the meta-analysis in comparison to those studies, which may provide additional insights.

Author Response

Point-by-Point Response to Reviewer’s Comments

We would like to sincerely thank the reviewers for their helpful recommendations. We have seriously considered all the comments and carefully revised the manuscript accordingly. Revisions are highlighted in yellow through the manuscript to indicate where changes have taken place. We feel that the quality of the manuscript has been significantly improved with these modifications and improvements based on the reviewers’ suggestions and comments. We hope our revision will lead to an acceptance of our manuscript for publication in International Journal of Environmental Research and Public Health.

In advance,

Kind regards

REVIEWER 3

Thank you very much for your contributions, with them you contribute to the improvement of our work. We will consider all your comments.

Regarding the feedback received, we will answer you below:

REVIEWER: One of my major concerns is the typos and double spaces throughout the manuscript (e.g., the first RMP in the abstract should actually be RPP), some major statistic results seem also wrongly presented (e.g., p=0.033 vs 0.33). All these reduce the credibility of the paper and compromise fluent reading.

AUTHORS: Thank you for your observation. The authors have carefully revised the manuscript to correct the typographical errors and double spaces.

REVIEWER: My second major concern is that the authors did not perform any subgroup analyses. This was perhaps because the heterogeneity analyses were not significant. However, even in the absence of heterogeneity within trials, subgroup analysis may provide important insights (see https://www.ncbi.nlm.nih.gov/pmc/articles/PMC2900281/; https://cegh.net/article/S2213-3984(18)30099-X/pdf). The authors may reconsider the differences between studies shown in, for instance, Figure 5 & 8, and check whether those differences could actually be attributed to subgroup differences. Relevant to this point, the authors mentioned some additional possible subgroup differences in lines 363-371, so the information of these features could not be included in Tables 2 & 3? Is there any possibility that some of these may cause any not-yet-discovered subgroup differences?

AUTHORS: Thank you for your recommendation. We find the idea of making comparisons by subgroups really interesting, although probably due to the number of articles found in the review (9 articles is no enough), the groups would be very small. Making the comparison by subgroups could make the results much more difficult, however we have include a table (Table 2) in which there are some of the independent variables that could be used to make the subgroups and the number of articles in each one of them and it is a really low number.

We have added a sentence in the text (line 355) referring to the difficulties of carrying out analyses among subgroups, taking into account the limitation of the small number of articles found.

REVIEWER: Another concern is that there seem to be studies that investigated the effect of listening to music using paradigms other than WAnT, since the authors mentioned that they excluded these studies (no problem at all); I think they may discuss the current findings of the meta-analysis in comparison to those studies, which may provide additional insights.

AUTHORS: Thank you for your observation. Although in the literature, there are more studies investigating the effect of music on predominantly aerobic than anaerobic activities, and taking into account that we find your contribution interesting; we have added a sentence referring to an article, in line 272, investigating the effect of music on vertical jumping as goal standart anaeorobic performace.

“Although all articles included in this section, evidenced that music could improve the experimental group response respect control group”,

Reviewer 4 Report

I appreciate the opportunity to review this systematic review and meta-analysis of experimental studies on the effects of listening to music on Wingate Anaerobic test performance.

 The genesis of this paper is an observation that although this is often used, there is no definitive opinion whether listening to music affects the effectiveness of the exercises. In their paper, the authors have conducted the meta-analysis of experimental trials in which the effects of during-listening to music tasks on 30-s Wingate anaerobic test (WAnT) performance absolute and relative peak power (APP and RMP), absolute and relative mean power (AMP and RMP), and fatigue index (FI) were tested.

I enjoyed reading the manuscript. I commend the authors for several strengths of their work, including clearly stated objectives and clinically relevant study question.

The authors conducted a comprehensive literature search (two different authors conducted the search for previous studies separately) and listed relevant databases (PUBMED/MEDLINE, Web of Science (WOS), Cochrane library, and Scopus). The authors also conducted a manual search through references of articles, abstracts? There were clear eligibility criteria for studies being chosen or rejected for the review, and the authors provided the characteristics of the studies listed. They also have used the appropriate statistical methods used to combine results, and the results were correctly displayed. The authors also used a satisfactory technique for assessing the risk of bias in individual studies.

The authors also provide a satisfactory discussion of some heterogeneity observed in the results of the review. As non-randomised studies were included variations in design and analysis might contribute to the heterogeneity

The authors admit that there is a minimal number of studies on this topic.

Have authors attempted collecting unpublished data?

Have they limited themselves to publication in English?

Author Response

Point-by-Point Response to Reviewer’s Comments

We would like to sincerely thank the reviewers for their helpful recommendations. We have seriously considered all the comments and carefully revised the manuscript accordingly. Revisions are highlighted in yellow through the manuscript to indicate where changes have taken place. We feel that the quality of the manuscript has been significantly improved with these modifications and improvements based on the reviewers’ suggestions and comments. We hope our revision will lead to an acceptance of our manuscript for publication in International Journal of Environmental Research and Public Health.

In advance,

Kind regards

REVIEWER 4

Thank you very much for your contributions, with them you contribute to the improvement of our work.

Regarding the feedback received, we will answer you below:

REVIEWER: The genesis of this paper is an observation that although this is often used, there is no definitive opinion whether listening to music affects the effectiveness of the exercises. In their paper, the authors have conducted the meta-analysis of experimental trials in which the effects of during-listening to music tasks on 30-s Wingate anaerobic test (WAnT) performance absolute and relative peak power (APP and RMP), absolute and relative mean power (AMP and RMP), and fatigue index (FI) were tested.

I enjoyed reading the manuscript. I commend the authors for several strengths of their work, including clearly stated objectives and clinically relevant study question.

The authors conducted a comprehensive literature search (two different authors conducted the search for previous studies separately) and listed relevant databases (PUBMED/MEDLINE, Web of Science (WOS), Cochrane library, and Scopus). The authors also conducted a manual search through references of articles, abstracts? There were clear eligibility criteria for studies being chosen or rejected for the review, and the authors provided the characteristics of the studies listed.

They also have used the appropriate statistical methods used to combine results, and the results were correctly displayed. The authors also used a satisfactory technique for assessing the risk of bias in individual studies.

The authors also provide a satisfactory discussion of some heterogeneity observed in the results of the review. As non-randomised studies were included variations in design and analysis might contribute to the heterogeneity

The authors admit that there is a minimal number of studies on this topic.

AUTHORS: Thank you for your great commentaries. The authors are very happy that you got a good impression of the manuscript. This is the main reason why we will consider all your comments

REVIEWER: Have authors attempted collecting unpublished data?

AUTHORS: Thank you for your interest. The authors did not attempt collecting unpublished data, we only have collected published data from relevant databases (PUBMED/MEDLINE, Web of Science (WOS), Cochrane library, and Scopus). To clarify this concern, we have added a new exclusion criterion: (V) studies no published in PUBMED/MEDLINE, web of Science (WOS), Cochrane library and Scopus.

Besides, the inclusion of data from unpublished studies can itself introduce bias. The studies that can be located may be an unrepresentative sample of all unpublished studies. Unpublished studies may be of lower methodological quality than published studies: a study of 60 meta-analyses that included published and unpublished trials found that unpublished trials were less likely to conceal intervention allocation adequately and to blind outcome assessments (Egger 2003). In contrast, Hopewell and colleagues found no difference in the quality of reporting of this information (Hopewell 2004).

REVIEWER: Have they limited themselves to publication in English?

AUTHORS: Thank you for your interest. One of our inclusion criteria was “(IV) in any language”. However, in order to ensure coverage of the best publications and as many as possible, all searches were conducted in English.

Reviewer 5 Report

This systematic review and meta-analysis attempted to evaluate the effect of listening to music on Wingate Anaerobic test (WAnT) performance.  The review found that based on meta-analysis of eight studies, there was no apparent beneficial effect of music on absolute peak power (APP), absolute mean power (AMP), or fatigue index (FI), but that relative peak power (RPP) and relative mean power (RMP) appeared to be increased with music.  Thus, the conclusion of the review stated that "listening to music during the WAnT could physiologically enhance relative anaerobic exercise performance, although reasons remain speculative."  The review is generally well conducted, but the conclusions are conflicting and unclear, and not necessarily supported by the data.  Several important methodological considerations should be addressed, and data interpretation should be reconsidered.

Major Comments:

1) The discrepancy between findings in absolute and relative peak and mean power are concerning.  If there is a true effect of music on performance, it should be evident in both absolute and relative power, because in this case body mass is not being affected by the intervention, thus relative power should only be affected by power output.  The fact that not all studies included in the analysis reported both absolute and mean power, or that insufficient data were available to successfully complete meta-analyses on both absolute and mean power is a major limitation of this study and weakens the conclusions that can be drawn from the analyses.  If a statistically significant effect can only be found when analyzing a subset of the studies (as is done in the relative power meta-analyses), then either the effect is not "real" (which is shown by including more studies), or if there is a "real" effect, it is obscured by adding additional studies to the analysis, perhaps by differences in study population, design, other variation introduced by including these studies, etc.)  Either way, this deserves a more detailed discussion in the paper, and should be acknowledged as a major limitation.

2) The wide variation between included studies in sample population, music selection, music rhythm, music volume, training status of subjects, experience of subjects with WAnT, time of day of testing, and many other factors which may affect the outcome of the tests limits the ability to draw meaningful conclusions from these studies.  For example, elite athletes tend to prefer associative, rather than dissociative, behaviors to optimize performance during exercise, whereas less experienced or recreational athletes tend to benefit from dissociative behaviors, such as listening to music.  This is not discussed in the paper.

3) Two studies which were included either didn't define the resistance setting, or used a resistance other than 7.5% of body mass, therefore these two studies should be excluded on the basis of not being conducted with the same standardized protocol - which the authors had pointed out as a strength of their analysis (Line 353), despite the fact that this was not true.

Minor Comments:

English language editing is required throughout.  Many sentences are missing articles, are improperly constructed, or otherwise difficult to interpret.

Abstract: MSMD not defined

Introduction: first two paragraphs are unnecessary - first paragraph is too general and unrelated to music and anaerobic performance, and second paragraph is a detailed description of the WAnT, which the audience of this paper should be familiar with.  Suggest omitting these paragraphs and devoting more discussion to the potential effects of music on performance, putative mechanisms, the nature of conflicting findings in previously published papers, and the current gap in understanding and need for the present review

Methods:

Line 106-107: Were CINAHL and SportDiscus searched?  These databases could contain other relevant records related to this topic

Line 112: Was the given search equation the ONLY one used, or did the authors use other combinations of terms?  If so, a comprehensive list of search strategies/equations/terms should be given so the search can be reproduced

Line 125: in Line 103 the "P" specified "adults" but in Line 125 it states that no filter was applied with regard to age.  Please clarify

Line 132: Exclusion criteria - studies which did not utilize a standard WAnT protocol should be excluded

Line 140-141: what is mean by "in addition to the results of APP and AMP that were related to participants' body mass (kg)"?  Isn't this simple RPP/RMP?

Line 177: risk of bias shows substantial amount of risk of bias with regard to selection bias and blinding of participants, as well as several studies with incomplete outcome data.  Given this, a quantitative assessment of study quality (such as PEDRO score) would be a helpful addition to the bias risk assessment.

Line 179: please define MSMD in statistical analysis section

Results:

Line 200: some numbers in text of this section (3.1, Main Search) do not match numbers in Figure 2 (PRISMA flow chart).  For example, the text states 294 articles "liked" (typo - this should be "linked" perhaps?) to the descriptors selected whereas the figure shows n=292 through the database search and 3 identified through additional searches.  Next, in lines 202-203 it states that 266 articles remained after duplicates were removed, but the figure shows n=267 after duplicates removed.  Please correct throughout with the correct numbers for consistency.

Table 3: why are there lines between Cutrufello and Hutchinson, between Hutchingson and Isik, and between Isik and Koc?  Are these studies meant to be separated for a reason?

Table 3: Why were Brohmer et al 2006 and Cutrufello et al 2019 included if they deviated from the standard 7.5% body mass resistance protocol for WAnT?

Figure 4: column heading says "Control" and "Control"; one of these should be corrected to "Experimental"

Figure 4: only one study had an SMD that was statistically significant - all others, despite a numerical trend towards improvement, were not statistically significant; please clarify and be specific in text so as not to mislead the reader to think that non-significant trends towards improvements were significant.

For all Forest plots, please provide total N and breakdown of sex (M/F) either in figure or in caption, since each analysis included a different number of studies

Line 227: relative values should be W/kg, not W/l

Discussion:

Lines 259-260: please clarify that these trends are "non-significant trends towards improvement in APP and AMP" to clarify these are not statistically significant.  The authors throughout use language that suggests these non-significant trends are meaningful when in fact, this is not a correct interpretation; please revise throughout

Lines 265-273: this paragraph is unnecessary, suggest omitting; moreover it is possible to calculate relative power in other tests, not just WAnT

Lines 274-279: this paragraph should be moved to the introduction

Lines 280-285: This disparity in findings is a major limitation of the study and should be discussed in greater detail, especially the difference in studies included in each analysis, and how this impacted the analysis.  Conducting meta-analyses on APP and AMP for ONLY the studies which have RPP and RMP could help show there is consistency between APP and RPP, as well as AMP and RMP, respectively; (since, hypothetically with no changes in body mass, the meta-analyses of APP and AMP should match the findings of the RPP and RMP analyses).  Given these significant findings from this subset of studies, and the fact that the effect becomes non-significant (albeit, nearly significant in the APP analysis) with the inclusion of more studies, suggests that perhaps either variability added by including more studies diminishes any potential "real" effect of the music intervention, or that the music intervention in fact does not exert a true effect.  This should be included in the discussion.

Lines 286-293: this section is not particularly relevant, suggest deleting

Line 332, and throughout: Some places Fatigue Index is abbreviated "FI" and others it's abbreviated "IF" please correct throughout to "FI" and check for consistency

Line 325: Other factors that impact fatigue index (training status, muscle fiber type distribution, etc.) should be discussed as potential influences and/or confounders on this outcome variable

Line 338: the relatively small sample size in all included studies (largest n=28) should be discussed as a limitation

Line 368: further discussion of the type, duration, tempo, volume, etc. of music selected and how this is inconsistent between studies and constitutes a limitation should be included.

Line 382: how a higher PP value be recommended with music if only RPP improved and APP did not?  The findings of this meta-analysis are too inconclusive and the data are too weak to provide and credible practical applications; suggest revising this paragraph to acknowledge there is a trend in RPP/RMP in the subset analysis, why that might be, and how these findings could potentially be applied either 1) if the effect truly is only in relative power (unlikely), or 2) if the findings of the subset analysis in fact do not represent the "true" effect (or lack of effect) of music on WAnT performance

Line 387: Conclusions - suggest revising according to above comments

Author Response

Point-by-Point Response to Reviewer’s Comments

We would like to sincerely thank the reviewers for their helpful recommendations. We have seriously considered all the comments and carefully revised the manuscript accordingly. Revisions are highlighted in yellow through the manuscript to indicate where changes have taken place. We feel that the quality of the manuscript has been significantly improved with these modifications and improvements based on the reviewers’ suggestions and comments. We hope our revision will lead to an acceptance of our manuscript for publication in International Journal of Environmental Research and Public Health.

In advance,

Kind regards

REVIEWER 5

Major Comments:

REVIEWER: 1) The discrepancy between findings in absolute and relative peak and mean power are concerning. If there is a true effect of music on performance, it should be evident in both absolute and relative power, because in this case body mass is not being affected by the intervention, thus relative power should only be affected by power output. The fact that not all studies included in the analysis reported both absolute and mean power, or that insufficient data were available to successfully complete meta-analyses on both absolute and mean power is a major limitation of this study and weakens the conclusions that can be drawn from the analyses.  If a statistically significant effect can only be found when analysing a subset of the studies (as is done in the relative power meta-analyses), then either the effect is not "real" (which is shown by including more studies), or if there is a "real" effect, it is obscured by adding additional studies to the analysis, perhaps by differences in study population, design, other variation introduced by including these studies, etc.)  Either way, this deserves a more detailed discussion in the paper, and should be acknowledged as a major limitation.

AUTHORS: Thank you for your observation. We know that this phenomenon is very difficult and unknown to explain. We included inside the discussion section (which we have modified), the next paragraph in order to explain these differences:Results of this systematic review and meta-analysis (only 9 articles for our update knowledge), suggest that listening to music may has a small effect on APP improvements and a trivial effect on increasing AMP. However, when the parameters were expressed in relation to body mass (RPP and RMP) effects were moderate. Although some psycho-physiological factors could explain these effects, (listening to different genres of music induces different psycho-physiological responses) (1), one explanatory variable of the variance of APP-RPP and MP-AMP performance could be the difference in studies included in each analysis, especially regarding to type of music. In that way, recent studies concluded that listening to music would enhance the speed of performance (related to anaerobic performance). Besides, music with a higher tempo (example Brohmer et al. 2006, with ac/dc music) increased the speed more (2). In this sense, only Isik et al. 2015 provided data regarding all power output outcomes (APP, RPP, AMP and RMP) (3). These results are consistent with other studies that have measured PP with other types of test such as vertical jump, in which the music improves some of the parameters related to the force in the vertical test (4).”.

Moreover, we have added this paragraph in the limitation section (line 343): “One of the most important limitations of this meta-analysis (9 articles) but is the update state of the art, is the fact that the founded data  in different ways, with some of the records provided being relative, absolute or both, which makes it difficult to conclude. Although the calculations of the meta-analysis have been carried out in a sub-group manner (main limitation) with the aim of clarifying the different tendencies, the way in which the data were found in each article determined the grouping of the data, and therefore it has not been possible to obtain unique and definitive meta-analytical calculations for each variable measured.”

REVIEWER: 2) The wide variation between included studies in sample population, music selection, music rhythm, music volume, training status of subjects, experience of subjects with WAnT, time of day of testing, and many other factors which may affect the outcome of the tests limits the ability to draw meaningful conclusions from these studies.  For example, elite athletes tend to prefer associative, rather than dissociative, behaviours to optimize performance during exercise, whereas less experienced or recreational athletes tend to benefit from dissociative behaviours, such as listening to music.  This is not discussed in the paper.

AUTHORS: When carrying out a meta-analysis, a number of articles with common characteristics are grouped together to try to give a conclusion about all of them, assuming that they have not carried out exactly the same methodology when conducting the research. These methodological differences may produce different results. In this meta-analysis we only try to draw a conclusion about which effect music could has (without specifying which type or also dependent on tastee), on Wingate performance test (without consider many test dependent variables that are applied differently in each research) in a generic way.

In that way it has reflected in the article, the mentioned limitations when the researches compared different methodologies. We have added the following paragraph: “Comparing the results obtained between subgroups (level of participants, type of music...) would have enriched the conclusions of the review but due to the small number of articles found, it is impossible to make a correct comparison. In addition, it is important to mention the small sample size of the selected studies, because none of the studies in the meta-analysis exceeds 30 participants, reducing the potential of the meta-analysis.”

REVIEWER: 3) Two studies which were included either didn't define the resistance setting, or used a resistance other than 7.5% of body mass, therefore these two studies should be excluded on the basis of not being conducted with the same standardized protocol - which the authors had pointed out as a strength of their analysis (Line 353), despite the fact that this was not true.

AUTHORS: Thank you for your observation. The reviewer should remember that although in 2 studies resistance is different from 7.5% of body mass, both the control and experimental groups have worked with the same protocol, so there is no bias in this regard that causes them to be due delete both studies. When calculating the meta-analysis, the difference between the two groups is used. Given that both groups belong to the same sample of participants with the same characteristics and that they have carried out the same work protocol, their elimination would be to lose 2 studies that meet the established inclusion / exclusion criteria.

Minor Comments:

REVIEWER: English language editing is required throughout.  Many sentences are missing articles, are improperly constructed, or otherwise difficult to interpret.

AUTHORS: Thank you for your observation. The text has been sent to a native reviewer to improve the English.

REVIEWER: Abstract: MSMD not defined

AUTHORS: Thank you for your observation. We have included the clarification of the concept in the abstract. Magnitude of Standardized Mean Differences. Line 35.

REVIEWER: Introduction: first two paragraphs are unnecessary - first paragraph is too general and unrelated to music and anaerobic performance, and second paragraph is a detailed description of the WAnT, which the audience of this paper should be familiar with.  Suggest omitting these paragraphs and devoting more discussion to the potential effects of music on performance, putative mechanisms, the nature of conflicting findings in previously published papers, and the current gap in understanding and need for the present review

AUTHORS: Thank you for your observation. We have reduced the contents not directly related to the meta-analysis as you say and we have changed the introduction and added content related to the music research carried out until now.

Methods:

REVIEWER: Line 106-107: Were CINAHL and SportDiscus searched?  These databases could contain other relevant records related to this topic

AUTHORS: Thank you for your observation. We have searched the Sportdiscus and CINAHL databases and added the name to the list provided in the article. No articles were found that could be added to the review.

REVIEWER: Line 112: Was the given search equation the ONLY one used, or did the authors use other combinations of terms?  If so, a comprehensive list of search strategies/equations/terms should be given so the search can be reproduced

AUTHORS: Thank you for your observation. After testing several search equation options, this was the final equation used. This option used was the one that appears in the manuscript due to the number and validity of the articles found with each of the evaluated options.

REVIEWER: Line 125: In Line 103, the "P" specified "adults" but in Line 125 it states that no filter was applied with regard to age.  Please clarify

AUTHORS: Thank you for your observation. We have included a clarification in the two parts of the text you comment on, in the P (line 100) and in line 121, referring to the fact that there have been no filters for age, but all the participants in the studies were more than 18 years old.

REVIEWER: Line 132: Exclusion criteria - studies which did not utilize a standard WAnT protocol should be excluded

AUTHORS: Thank you for your observation. The initial WAnT protocol was designed to assess sedentary participants, which has been the reason for study and discussion in the following years, since many of the guidelines given in its protocol did not allow trained participants to achieve maximum performance.

The standard protocol of the WAnT is not respected by almost all the research, since the same warm-up is not performed (the original protocol recommends a warm-up with changes in intensity), or the test is started from a stationary position (in the original protocol it is recommended to start the test from about 60-70 pedaling cycles per minute), or the subject's position during the test is not controlled (whether he can stand up on the bike or not).

The resistance provided by the bike during the test is a special case, because although the initial protocol was designed with the 7.5% you mention, several investigations have studied the adequacy of this measure to obtain the best performance and conclude that in non-sedentary people the resistance provided by the bike to obtain the best performance should be greater than 7.5%. Moreover, many studies indicate that defining resistance by the participants's body mass is not the best way to do it and it should be done taking into account the lean body mass.

https://www.ncbi.nlm.nih.gov/pubmed/25849068

https://www.thieme-connect.com/products/ejournals/html/10.1055/a-1114-6206

https://www.researchgate.net/publication/27794614_The_Load_of_the_Wingate_Test_According_to_the_Body_Weight_or_Lean_Body_Mass

https://www.ncbi.nlm.nih.gov/pubmed/4008145

https://www.ncbi.nlm.nih.gov/pubmed/6685039

Even so, our objective is not to assess if the gold standard WAnT protocol is well or badly implemented in the research, but to assess if they have carried out a methodology that allows them to compare the effect of music on WAnT performance, with one resistance or another, with one music or another or with a biomechanical adjustment of the bicycle before the test or another.

REVIEWER: Line 140-141: what is mean by "in addition to the results of APP and AMP that were related to participants' body mass (kg)"?  Isn't this simple RPP/RMP?

AUTHORS: Thank you for your observation. We have modified the text to correct your comments. With the changes it has become line 135.

REVIEWER: Line 177: risk of bias shows substantial amount of risk of bias with regard to selection bias and blinding of participants, as well as several studies with incomplete outcome data.  Given this, a quantitative assessment of study quality (such as PEDRO score) would be a helpful addition to the bias risk assessment.

AUTHORS: Thank you for your observation. Following your recommendation and in order to reduce the risk of bias with regard to selection bias and blinding of participants, we have added a quantitative assessment of the quality of the studies with PEDro's score.

Besides, we consider that in an investigation of this characteristics, it is impossible to blind the participants, since it is required to listen to music and you will always be aware of knowing if the participant is listening to music or not.

REVIEWER: Line 179: please define MSMD in statistical analysis section

AUTHORS: Thank you for your interest. We have included the clarification of the concept. Magnitude of Standardized Mean Differences in line 168.

Results:

REVIEWER: Line 200: some numbers in text of this section (3.1, Main Search) do not match numbers in Figure 2 (PRISMA flow chart).  For example, the text states 294 articles "liked" (typo - this should be "linked" perhaps?) to the descriptors selected whereas the figure shows n=292 through the database search and 3 identified through additional searches.  Next, in lines 202-203 it states that 266 articles remained after duplicates were removed, but the figure shows n=267 after duplicates removed.  Please correct throughout with the correct numbers for consistency.

AUTHORS: Thank you for your observation. The authors have reviewed this section to avoid mistakes.

REVIEWER: Table 3: why are there lines between Cutrufello and Hutchinson, between Hutchingson and Isik, and between Isik and Koc?  Are these studies meant to be separated for a reason?

AUTHORS: Thank you for your observation. For some reason of format, the other lines had disappeared. We have already included all the lines between articles.

REVIEWER: Table 3: Why were Brohmer et al 2006 and Cutrufello et al 2019 included if they deviated from the standard 7.5% body mass resistance protocol for WAnT?

AUTHORS: Thank you for your observation. Due to the small number of items found, a comparison by subgroups (different sex, type of music or different resistance used on the bicycle) could not be made.

As we included in the limitations in line 363, the most methodologically correct would has been to differentiate the articles that have used different resistances, however, we believe that these studies have compared the performance of a group of participants with and without music and with the same protocol in both cases, so they can be included equally in the review and meta-analysis.

REVIEWER: Figure 4: column heading says "Control" and "Control"; one of these should be corrected to "Experimental"

AUTHORS: Thank you for your recommendation. We have modified Figure 4 by correcting the error that you are mentioning.

REVIEWER: Figure 4: only one study had an SMD that was statistically significant - all others, despite a numerical trend towards improvement, were not statistically significant; please clarify and be specific in text so as not to mislead the reader to think that non-significant trends towards improvements were significant.

AUTHORS: Thank you for your observation. To avoid misunderstandings, authors have changed this sentence: “Although all the articles included in this section evidenced that music could improvement in the experimental group versus control group, the results of the meta-analysis indicated that listening to music did not statistically improve the absolute APP(SMD: 0.20; 95% CI: -0.03 to 0.43; MSMD, small; I2, 0%; p= 0.09).”

REVIEWER: For all Forest plots, please provide total N and breakdown of sex (M/F) either in figure or in caption, since each analysis included a different number of studies

AUTHORS: Thank you for your recommendation. The N of experimental and control groups are included in the forest plots (columns 4 and 7). However, the software used to do it allows us to differentiate between males and females. However, in table 3 where the sample of each study is described, it can be seen what distribution of M/F each study had.

REVIEWER: Line 227: relative values should be W/kg, not W/l

AUTHORS: Thank you for your observation. We have corrected this typo (line 216).

Discussion:

REVIEWER: Lines 259-260: please clarify that these trends are "non-significant trends towards improvement in APP and AMP" to clarify these are not statistically significant.  The authors throughout use language that suggests these non-significant trends are meaningful when in fact, this is not a correct interpretation; please revise throughout

AUTHORS: Thank you for your observation. To avoid misunderstandings, authors have changed this sentence: “Moreover, when the athletes listen to music during WAnT, the results of the meta-analysis showed no statistical difference to interpreter that listening to music improved APP and AMP. However, the results showed that listening to music during WAnT statistically improved RPP and RMP.”

REVIEWER: Lines 265-273: this paragraph is unnecessary, suggest omitting; moreover, it is possible to calculate relative power in other tests, not just WAnT

AUTHORS: Thank you for your recommendation. The authors have removed this paragraph.

REVIEWER: Lines 274-279: this paragraph should be moved to the introduction

AUTHORS: Thank you for your recommendation. We have included the paragraph in the introduction.

REVIEWER: Lines 280-285: This disparity in findings is a major limitation of the study and should be discussed in greater detail, especially the difference in studies included in each analysis, and how this impacted the analysis.  Conducting meta-analyses on APP and AMP for ONLY the studies which have RPP and RMP could help show there is consistency between APP and RPP, as well as AMP and RMP, respectively; (since, hypothetically with no changes in body mass, the meta-analyses of APP and AMP should match the findings of the RPP and RMP analyses).  Given these significant findings from this subset of studies, and the fact that the effect becomes non-significant (albeit, nearly significant in the APP analysis) with the inclusion of more studies, suggests that perhaps either variability added by including more studies diminishes any potential "real" effect of the music intervention, or that the music intervention in fact does not exert a true effect.  This should be included in the discussion.

AUTHORS: Thank you for your observation. The authors have added a paragraph to the discussion. “The meta-analysis performed by subgroups with APP, MPA, RPP and RMP should help to demonstrate that there is consistency between the results of the subgroups (APP and RPP as well as MPA and RMP), however, and given that body mass is inherent to the body itself, the results of the APP and MPA meta-analyses should be consistent with the results of RPP and RMP. Given these significant findings from this subset of studies, and the fact that the effect becomes non-significant ( albeit, almost significant in the PPP analysis) with the inclusion of more studies, suggests that perhaps either the variability added by the inclusion of more studies decreases any potential "real" effect of the music intervention, or that the music intervention does not in fact exert a true effect. However, it is important to take into account the low heterogeneity (0%) of the sample, so it is considered to have been conducted under similar conditions with similar subjects – in other words, the only difference between studies is their power to detect the outcome of interest [56]”.

REVIEWER: Lines 286-293: this section is not particularly relevant, suggest deleting

AUTHORS: Thank you for your recommendation. The authors haver removed this paragraph.

REVIEWER: Line 332, and throughout: Some places Fatigue Index is abbreviated "FI" and others it's abbreviated "IF" please correct throughout to "FI" and check for consistency

AUTHORS: Thank you for your observation. We have revised all the related abbreviations in the text.

REVIEWER: Line 325: Other factors that impact fatigue index (training status, muscle fiber type distribution, etc.) should be discussed as potential influences and/or confounders on this outcome variable

AUTHORS: Thank you for your observation. The authors have added a paragraph to the discussion. “It is important to take into account that the FI is very sensible to training effects and to the subject's sport specialty (Calbet 1997), besides depending on the type of fibers of the participant, since a higher percentage of fibers facilitates a higher PP but with a higher FI, due to the difficulty to maintain those high power values with a lower aerobic capacity of the fast fibers (Janot 2015)”.

REVIEWER: Line 338: the relatively small sample size in all included studies (largest n=28) should be discussed as a limitation

AUTHORS: Thank you for your observation. In order to reflect the idea we have added a sentence on the limitations of the study: “It is important to mention the small sample size of the selected studies, because none of the studies in the meta-analysis exceeds 30 participants, reducing the potential of the meta-analysis.”

REVIEWER: Line 368: further discussion of the type, duration, tempo, volume, etc. of music selected and how this is inconsistent between studies and constitutes a limitation should be included.

AUTHORS: Thank you for your recommendation. The authors haver added this paragraph.  “Those related to WAnT as they can be: the type of warm-up before the test [66] as a warm-up with changes of pace may help to improve performance later on in the test; resistance provided by the cicloergometer [67] as a resistance that is too low, such as the recommended 7.5% of de body mass, may not help to achieve maximum performance; the time of day when the test is done [65], in the evening, may increase test performance; the position adopted by the subject in the cicloergometer during the test [68], the change of position may help to improve performance; the geometry of the bicycle [69] which, although it seems to have no influence, is not very well studied. In relation to the music used: the volume at which the music is played can influence performance [26], and a higher volume can be more motivating; the participants' previous experiences of the participants with music [10], because a song can recall moments and emotions for one subject and not for another; whether the music is to their favourite or not [26,32], which can generate greater motivation;”

REVIEWER: Line 382: how a higher PP value be recommended with music if only RPP improved and APP did not?  The findings of this meta-analysis are too inconclusive and the data are too weak to provide and credible practical applications; suggest revising this paragraph to acknowledge there is a trend in RPP/RMP in the subset analysis, why that might be, and how these findings could potentially be applied either 1) if the effect truly is only in relative power (unlikely), or 2) if the findings of the subset analysis in fact do not represent the "true" effect (or lack of effect) of music on WAnT performance

AUTHORS: Thank you for your recommendation. The authors have removed the affirmative tone from the sentence, making clear the possible effect of music on performance in WAnT with the words in yellow.

“It is important to take into account that this potential effect could not always interest since if what we are looking for is to maintain great power that is as constant as possible (a low FI), the music may not like us, while if the main aim is a high PP value, then it seems that it might be interested in that possible effect.(5)”

REVIEWER: Line 387: Conclusions - suggest revising according to above comments

AUTHORS: Thank you for your recommendation. We have modified the conclusions in accordance with the corrections made.

“Bases on the results of this systematic review and meta-analysis, it is clear that listening to music during WAnT does not improve the ana more than 18 years old aerobic performance more than 18 years old. Although there is a tendency to improve on results obtained in most studies, these differences are not statistically significant. For our knowledge, the studies published on this topic are scarce and future research could confirm these findings. The results reported in this systematic review and meta-analysis although show a trend towards improvement, we cannot confirm it yet. Although it is still subject to speculation, it is believed that the effects of music could derive from other unknown psychophysiological factors.”

Round 2

Reviewer 1 Report

Thank you for making the contribution of the new data and the changes in the structure of the manuscript.

Author Response

We would like to sincerely thank again the reviewer for his/her helpful recommendations. 

Reviewer 2 Report

The authors provided a novel version, addressing all the misleading points.

English has been improved and novel references have been added

The conclusion is now fine showing the real limit of this type of study.

The paper can be considered for publication. 

Author Response

(The authors gave the same response as above.)

Reviewer 3 Report

Thank the authors for making the revisions. Regarding Table 4, 1), the expressions of b/min, beats/min, and bpm are inconsistent, please also check similar expressions throughout the manuscript; 2) please specify the criterion of being statistically significant in this table (column "main conclusions").

Author Response

Point-by-Point Response to Reviewer’s Comments

We would like to sincerely thank again the reviewer for his/her helpful recommendations. We have seriously considered all the comments and carefully revised the manuscript accordingly. Revisions are highlighted in yellow through the manuscript to indicate where changes have taken place. We feel that the quality of the manuscript has been significantly improved with these modifications and improvements based on the reviewers’ suggestions and comments. We hope our revision will lead to an acceptance of our manuscript for publication in International Journal of Environmental Research and Public Health.

In advance,

King regards

REVIEWER 3

Thank the authors for making the revisions.

REVIEWER: Regarding Table 4, 1), the expressions of b/min, beats/min, and bpm are inconsistent, please also check similar expressions throughout the manuscript

AUTHORS: Thank you for your observation. The thermology in table number 4 and during the text has been corrected.

REVIEWER: 2) please specify the criterion of being statistically significant in this table (column "main conclusions").

AUTHORS: Thank you for your observation. The authors have added in the footer of table the criterion of being statistically significant in the column "main conclusions": “↑ Statistically significant increase (p <0.05); ↔ no statistical significance changes (p >0.05); ↓: statistically significant decrease (p <0.05).”

Reviewer 5 Report

Thank you to the authors for this thorough revision which adequately addresses the majority of my previous comments.  Some additional comments are provided below:

1) Please define all abbreviations at first use in the main text body (i.e., if it's defined in the abstract, please re-define in the main body text, as the abstract is considered separate from the main body); for example, APP, AMP, RMP, etc. in the Introduction section - abbreviations are used in Line 75, but not defined until the following paragraph in Lines 80-82.

2) Methods: Please clarify that this was the only search equation used (as described in Author's reply)

3) Methods: Please clarify if this systematic review and meta-analysis was prospectively registered in PROSPERO, and if so, include registration number

4) Methods: please add a section describing methods for quality assessment (e.g., evaluation of bias and PEDRO scoring); this is somewhat given in the Results, but the Results should report the findings of this assessment whereas the description of how this was done should be reported in the Methods

5) Suggest including justification of including studies that deviated from standard 7.5% body mass workload - this could easily be done by incorporating the references and discussion from the Author's reply to the original comment on Line 132.

6) Practical Applications section: suggest making explicit statement that while music may be of interest for increasing PP, based on current meta-analysis, there is insufficient evidence to provide a recommendation

Author Response

Point-by-Point Response to Reviewer’s Comments

We would like to sincerely thank again the reviewer for his/her helpful recommendations. We have seriously considered all the comments and carefully revised the manuscript accordingly. Revisions are highlighted in yellow through the manuscript to indicate where changes have taken place. We feel that the quality of the manuscript has been significantly improved with these modifications and improvements based on the reviewers’ suggestions and comments. We hope our revision will lead to an acceptance of our manuscript for publication in International Journal of Environmental Research and Public Health.

In advance,

King regards

REVIEWER 5

REVIEWER: Thank you to the authors for this thorough revision which adequately addresses the majority of my previous comments.  Some additional comments are provided below:

AUTHORS: Thank you for your words. We have tried we have tried to answer your insightful comments.

REVIEWER: 1) Please define all abbreviations at first use in the main text body (i.e., if it's defined in the abstract, please re-define in the main body text, as the abstract is considered separate from the main body); for example, APP, AMP, RMP, etc. in the Introduction section - abbreviations are used in Line 75, but not defined until the following paragraph in Lines 80-82.

AUTHORS: Thank you for your observation. The authors haver defined all abbreviations at first use.

REVIEWER: 2) Methods: Please clarify that this was the only search equation used (as described in Author's reply)

AUTHORS: Thank you for your observation. The text has been modified including the following sentence in line 105 “with the following unique search equation”.

REVIEWER: 3) Methods: Please clarify if this systematic review and meta-analysis was prospectively registered in PROSPERO, and if so, include registration number

AUTHORS: Thank you for your observation. Thanks for the recommendation. As we commented to reviewer number 1, it was not possible to record this systematic review and meta-analysis in the PROSPERO database at this time is unable to accept our application given that the outcomes that we analysed were on athletic performance related and they are not on human health related:  

PROSPERO QUESTION: “Does this review have at least one outcome directly related to human health or is it a methodology review that has a clear link to human health?

PROSPERO ANSWER: Sorry - your protocol is not eligible for inclusion in PROSPERO. We hope that this screening process has saved you the time that might be spent preparing a submission that would be rejected and that you will register future systematic reviews in PROSPERO. Further information about PROSPERO scope and eligibility can be found in our guidance document.”

However, although this systematic review and meta-analysis are not registered in PROSPERO, we have followed the PRISMA statement and the same structure that previous systematic review and meta-analysis published recently by or research group in other Q1 journal without PROSPERO registration.

REVIEWER: 4) Methods: please add a section describing methods for quality assessment (e.g., evaluation of bias and PEDRO scoring); this is somewhat given in the Results, but the Results should report the findings of this assessment whereas the description of how this was done should be reported in the Methods

AUTHORS: Thank you for your observation. The authors have created a new section in methods section. On the one hand, we have translated some information of results section: “Two separate authors evaluated quality in terms of the methodology used together with any risk of bias (A.C-B and J.M-A), with any lack of consensus being subject to third-party assessment (D.M-J), pursuant to Cochrane Collaboration Guidelines [46]. Items on the list were broken down into seven distinct areas: random sequence generation (section bias), allocation concealment (section bias), the fact of participants and staff being blinded (performance bias), blinding of outcome assessment (detection bias), incomplete outcome data (attrition bias) and selective reporting (reporting bias), as well as other types of bias (design-specific risks of bias, baseline imbalance, blocked randomization in unblinded trials and differential diagnostic activity). In cases where criteria for low risk of bias were fulfilled, these were deemed to be “low” (plausible bias unlikely to modify results to a major extent) or “high” in cases where criteria for high risk of bias were fulfilled (plausible bias that may reduce confidence in results to a great extent). In cases where it was unclear whether there was any risk of bias, it was deemed “unclear” accordingly (plausible bias that raises some doubts about the results).”

On the other hand, we have added information about PEDRO scoring: “Moreover, to determine the quality of the evidence, the authors reviewed the considered articles and provided PEDro (Physiotherapy Evidence Database) scores for each article. Only studies with PEDro scores of 4 or higher were considered for the systematic review. According to Maher et al., the PEDro scale is an 11-item scale designed for rating methodological quality of randomized control trials [53]. Each satisfied item (except for item 1) contributes one point to the total PEDro score (0-10 points) [53]. The PEDro scores were extracted from the PEDro database. If a study had not been entered into the database and scored, it was reviewed and scored by an experienced PEDro rater.”

REVIEWER: 5) Suggest including justification of including studies that deviated from standard 7.5% body mass workload - this could easily be done by incorporating the references and discussion from the Author's reply to the original comment on Line 132.

AUTHORS: Thank you for your observation. The authors have included in Inclusion and Exclusion criteria section: “In addition, other inclusion criteria was to include studies in which both the experimental group and the control used the same resistance parameter although it deviated from standard of 7.5% of body mass workload. In this sense, even if the initial protocol was designed with the 7.5% of the body mass, several investigations have studied the adequacy of this measure to obtain the best performance and concluded that in non-sedentary people the resistance provided by the bike to obtain the best performance should be greater than 7.5% [48–51]. Moreover, some studies indicated that defining resistance by the participants’ body mass is not the best parameter to apply it and it should be done taking into account the lean body mass [52].”

REVIEWER: 6) Practical Applications section: suggest making explicit statement that while music may be of interest for increasing PP, based on current meta-analysis, there is insufficient evidence to provide a recommendation

AUTHORS: Thank you for your observation. The authors have included this sentence in plactical applications section: “However, while music may be of interest for increasing PP, based on current meta-analysis, there is insufficient evidence to provide a recommendation.”